# Serum Mac-2 Binding Protein Levels Associate with Metabolic Parameters and Predict Liver Fibrosis Progression in Subjects with Fatty Liver Disease: A 7-Year Longitudinal Study

**DOI:** 10.3390/nu12061770

**Published:** 2020-06-12

**Authors:** Yoshihiro Kamada, Koichi Morishita, Masahiro Koseki, Mayu Nishida, Tatsuya Asuka, Yukiko Naito, Makoto Yamada, Shinji Takamatsu, Yasushi Sakata, Tetsuo Takehara, Eiji Miyoshi

**Affiliations:** 1Department of Molecular Biochemistry and Clinical Investigation, Osaka University Graduate School of Medicine, Suita, Osaka 565-0871, Japan; ykamada@gh.med.osaka-u.ac.jp (Y.K.); a.xxivk.m@gmail.com (K.M.); nmayu.1119@gmail.com (M.N.); tasuka25@gmail.com (T.A.); rin.hana.ren@gmail.com (Y.N.); shinjit@sahs.med.osaka-u.ac.jp (S.T.); 2Department of Gastroenterology and Hepatology, Osaka University Graduate School of Medicine, Suita, Osaka 565-0871, Japan; takehara@gh.med.osaka-u.ac.jp; 3Department of Cardiovascular Medicine, Osaka University Graduate School of Medicine, Suita, Osaka 565-0871, Japan; koseki@cardiology.med.osaka-u.ac.jp (M.K.); yasushisk@cardiology.med.osaka-u.ac.jp (Y.S.); 4aMs NEW OTANI CLINIC, Osaka 540-0001, Japan; kazuaki.sakata@ams-group.jp

**Keywords:** Mac-2 binding protein, fatty liver, metabolic syndrome, liver fibrosis, biomarker, longitudinal study

## Abstract

*Background:* Mac-2 binding protein (M2BP) is a highly glycosylated secreted glycoprotein that is involved in immune defense and regulation. Our cross-sectional studies indicated that serum M2BP was a useful liver fibrosis biomarker for nonalcoholic fatty liver disease (NAFLD). In this study, we conducted a 7-year longitudinal study to investigate the significance of serum M2BP levels (baseline and at 7-year follow-up) and their relationships with other metabolic parameters of fatty liver disease. *Methods:* We enrolled 715 study subjects (521 male and 194 female) during health examinations. Study subjects received blood sampling tests and abdominal ultrasound tests at baseline and follow-up. *Results:* Univariate analyses demonstrated that serum M2BP levels were significantly correlated with various parameters related to metabolic risk (body mass index (BMI), systolic blood pressure, triglyceride, high density lipoprotein (HDL)-cholesterol) and metabolic syndrome diseases (obesity, hypertension, dyslipidemia, diabetes mellitus, fatty liver (FL)). Multiple logistic regression analyses demonstrated that BMI and FL were independent determinants for serum M2BP levels. Baseline serum M2BP levels were significant independent determinants for changes in platelet count, Fibrosis-4 (FIB4) index, and NAFLD fibrosis score. Higher serum M2BP levels (>1.80 μg/mL) strongly correlated with changes in the FIB4-index. *Conclusions*: The results of this study suggest that changes in serum M2BP levels reflect changes in specific metabolic disease-related parameters, and baseline serum M2BP levels could predict changes in liver fibrosis.

## 1. Introduction

Nonalcoholic fatty liver disease (NAFLD) is one of the most common causes of chronic liver disease and a growing medical problem worldwide [1]. A wide spectrum of hepatic histological changes occur in NAFLD patients, ranging from the generally nonprogressive nonalcoholic fatty liver to nonalcoholic steatohepatitis (NASH). It is important to assess the degree of liver fibrosis during the clinical progression of NAFLD to predict disease progression and formulate therapeutic management decisions [2,3]. Liver biopsy remains the gold standard for assessing liver fibrosis [4,5]. However, liver biopsy has significant limitations such as pain, risk of severe complications, sampling error [6], cost [7], and patient unwillingness to undergo invasive testing. A recent study reported that liver fibrosis was independently associated with long-term outcome in NAFLD patients [8]. Therefore, there is an urgent need for a reliable and noninvasive test that accurately assesses the degree of liver fibrosis. Recently, novel noninvasive approaches such as transient elastography (e.g., FibroScan, acoustic radiation force impulse), magnetic resonance elastography (MRE), and various scoring systems (e.g., Fibrosis (FIB)-4 index, NAFLD fibrosis score) can be used to measure the liver fibrosis severity in patients with chronic liver disease [9,10,11,12]. However, distinguishing early liver fibrosis from non-fibrotic liver is difficult using transient elastography. Various scoring systems and biomarkers for measuring liver fibrosis severity are available [13,14,15], but few individual systems or biomarkers have clinical validity. Therefore, there is still the need for a reliable liver fibrosis biomarker.

Mac-2 binding protein (M2BP) is a glycoprotein that contains seven potential *N*-glycosylation sites [16,17]. We previously identified M2BP as one of the major fucosylated glycoproteins that were secreted from the HuCCT-1 liver bile duct cancer cell line [18]. Serum M2BP concentrations increase in patients with various cancers (e.g., pancreatic, breast, and lung cancer), viral hepatitis, and autoimmune disease [16]. M2BP is essentially undetectable in normal liver, but is easily detected in hepatocytes from patients with chronic hepatitis type C (CHC) during the progression of liver fibrosis [19,20]. Previously, we developed an enzyme-linked immunosorbent assay (ELISA) kit to measure serum total M2BP levels in human [18].

We recently observed that serum M2BP levels could be used to predict the histological severity of hepatic fibrosis in NAFLD patients [18,21]. *Wisteria floribunda* agglutinin (WFA)-positive M2BP (WFA^+^-M2BP), also known as M2BPGi (Mac-2 Binding Protein Glycosylation isomer), is a novel serum fibrosis biomarker for CHC [22]. This biomarker distinguishes the glycan structure of WFA-detectable M2BP; it was developed using a glycan-based immunoassay for assessing liver fibrosis severity in CHC patients [22]. WFA^+^-M2BP also can be used for assessing the liver fibrosis stage in NAFLD patients [23]. Our recent meta-analysis suggested that serum WFA^+^-M2BP was a reliable biomarker that detected advanced fibrosis in various chronic liver diseases [24]. However, the liver fibrosis prediction abilities for the other chronic liver diseases (NAFLD, chronic hepatitis type B, primary biliary cirrhosis, and autoimmune hepatitis) are relatively lower than that for CHC [22,23,25,26,27]. Previously, our study also revealed that compared with WFA^+^-M2BP, total M2BP measured by our developed ELISA had a greater ability to predict NAFLD fibrosis stage [28]. In this study, we measured serum total M2BP levels in our study subjects.

These combined results indicate that serum M2BP levels are useful liver fibrosis biomarkers. However, the relationships among serum M2BP levels, metabolic parameters, and liver fibrosis progression have not been investigated in a longitudinal study. The objective of this study was to examine the relationships among changes in serum M2BP levels, changes in metabolic parameters, and changes in liver fibrosis markers using a 7-year follow-up study. Recent studies reported that alcohol and obesity synergistically sensitize liver damage [29,30]. The concept of metabolic dysfunction–associated fatty liver disease (MAFLD) was proposed recently [31]. The criteria for MAFLD regardless of alcohol consumption are based on evidence of fatty liver along with one of the following three parameters: overweight/obesity, presence of type 2 diabetes mellitus (DM), or evidence of metabolic dysregulation. Our present study investigated MAFLD subjects and did not exclude subjects with a history of alcohol abuse.

## 2. Patients and Methods

### 2.1. Ethical Committee Approval

The research and informed consent protocols were approved for use in a multicenter study by the institutional review boards at Osaka University Hospital and aMs NEW OTANI CLINIC. Written informed consent was obtained from each subject at time of enrollment at each institute. The study was conducted in accordance with the Helsinki Declaration.

### 2.2. Subjects in Medical Health Check-Ups

A total of 2167 individuals who underwent a health examination at aMs NEW OTANI CLINIC (Osaka, Japan) from 2009 to 2011 were initially recruited into the study, and 806 of these subjects received a follow-up health examination after 7 years. No specific inclusion criteria were applied. The following exclusion criteria were applied: history of hepatic disease such as CHC or concurrent active hepatitis B (seropositive for hepatitis B surface antigen), autoimmune hepatitis, primary biliary cholangitis, sclerosing cholangitis, hemochromatosis, α1-antitrypsin deficiency, Wilson’s disease, and hepatic injury caused by substance abuse excluding alcohol consumption. Alcohol consumption (g/week) was calculated based on the self-response questionnaire sheets at medical check-ups. Subjects receiving anticoagulant therapy, which could affect platelet measurements, were also excluded. A total of 715 subjects (521 male and 194 female) received an abdominal ultrasound test. The diagnosis of fatty liver was based on the results of the abdominal ultrasound examination performed by trained technicians. A fatty liver was defined as a liver parenchyma with an echogenicity greater than that of the kidney cortex, the presence of vascular blurring, and deep attenuation of the ultrasound signal [32,33]. Blood serum was collected from each subject during the health examination and was stored at −80 °C until analysis.

### 2.3. Anthropometric and Laboratory Evaluation

Anthropometric variables (height and weight) were measured while each subject was in the standing position. Body mass index (BMI) was calculated as weight (kg) divided by the square of height in meters (m^2^). Systolic and diastolic blood pressure (SBP and DBP) values were measured (to the nearest mm Hg) while each subject was in the sitting position. Serum biochemical variables [aspartate aminotransferase (AST), alanine aminotransferase (ALT), γ-glutamyltransferase (GGT), total bilirubin (T-Bil), creatinine, choline esterase (CHE), total cholesterol (T-Chol), triglyceride (TG), high density lipoprotein cholesterol (HDL-C), low density lipoprotein cholesterol (LDL-C), uric acid, iron, fasting blood glucose (FBG), albumin, and platelet count] were measured using a conventional automated blood analyzer. The FIB4-index (based on subject age, serum AST and ALT levels, and platelet counts) was calculated for each subject as reported previously [age × AST (U/L)/Platelet count (×10^9^/L)/√ALT (U/L)] [34,35]. Impaired fasting glucose (IFG) was defined as an FBG of 110–125 mg/dL. The NAFLD fibrosis score (NFS) was calculated for each of the subjects as previously reported [1.675 + 0.037 × age (years) + 0.094 × BMI (kg/m^2^) + 1.13 × IFG/DM (yes = 1, no = 0) + 0.99 × AST/ALT − 0.013 × platelet count (×10^9^/L) − 0.66 × albumin (g/dL)] [12]. We used an ELISA kit (Immuno-Biological Laboratory Co. Ltd., Fujioka, Japan, code # 27362) to measure serum M2BP [18]. Changes in various parameters were defined as follows: ((parameter data at follow-up) − (parameter data at baseline)).

### 2.4. Diagnostic Criteria for Metabolic Syndrome-Related Diseases

Diagnostic criteria for metabolic syndrome-related diseases were as follows: obesity [BMI ≥ 25 (*n* = 310, 43.3%)]; hypertension [history of drug use, SBP ≥ 130, or DBP ≥ 85 (*n* = 250, 34.9%)]; dyslipidemia [history of drug use, TG ≥ 150, and/or HDL-C < 40 (*n* = 279, 39.0%)]; and diabetes mellitus [history of drug use, FBG ≥ 126, or HbA1c ≥ 6.5% (*n* = 428, 59.8%)]. There were no significant differences among subjects with or without drug use (antihypertensive drug, dyslipidemia drug) in AST and ALT values.

### 2.5. Statistical Analysis

Statistical analyses were performed using JMP Pro 14.0 software (SAS Institute Inc., Cary, NC, USA). Results were expressed as mean ± standard deviation (SD). The statistical analysis included descriptive statistics, analysis of variance, the Wilcoxon test, and the Pearson test. Chi-square tests were used for categorical factors. The AST, ALT, GGT, TG, FBG, HbA1c, M2BP, and FIB4-index values did not display Gaussian distributions; therefore, these parameters were common log-transformed before analysis. Multivariate logistic regression analyses were performed to identify significant determinants. Differences were considered as statistically significant at *p* < 0.05.

## 3. Results

### 3.1. Characteristics of the Study Participants

The results of the clinical and biochemical analyses of individuals in the study population are presented in Table 1. Among the 715 study subjects, 442 were diagnosed with fatty liver disease at baseline using abdominal ultrasound, and 423 subjects were diagnosed with fatty liver disease at follow-up. Among 442 fatty liver subjects at baseline, 382 subjects (86.4%) were still diagnosed with fatty liver at follow-up. The ratio of subjects with fatty liver disease at follow-up was significantly lower than that at baseline. The ratio of males in the group with fatty liver disease (FL+) was greater than the ratio of males in groups without fatty liver disease (FL−) at both baseline and follow-up. BMI was greater in subjects with fatty liver disease. Alcohol consumption (g/week) was not different between (FL+) and (FL−) at baseline, but significantly higher in (FL+) than in (FL−) at follow-up. The serum levels of AST, ALT, GGT, albumin, CHE, TG, LDL-C, uric acid, and FBG were significantly higher in FL+ subjects than in FL− subjects at both baseline and follow-up. The FIB4 index was lower in FL+ subjects than in FL− subjects. The NFS was not different between (FL+) and (FL−) at baseline, but significantly higher in (FL+) than in (FL−) at follow-up. Serum HDL-C levels were lower in FL+ subjects. Serum M2BP levels were higher in FL+ subjects at both baseline and follow-up.

### 3.2. Serum M2BP Levels were Significantly Correlated with Liver Enzymes and Metabolic-Related Variables

The results of Pearson’s correlations analyses between serum M2BP levels and other liver and metabolic parameters at baseline are summarized in Table 2. Serum M2BP levels were significantly and positively correlated with BMI, SBP, AST, ALT, GGT, CHE, TG, T-Chol, LDL-C, uric acid, FBG, and HbA1c, whereas they were negatively correlated with T-Bil and HDL-C. These data suggested that serum M2BP levels were higher in subjects with metabolic disorder condition and/or liver dysfunction.

### 3.3. Among Metabolic Syndrome-Related Diseases, Fatty Liver Disease Was an Independent Determinant for Serum M2BP Levels

We investigated the relationships between serum M2BP levels and metabolic syndrome-related diseases by comparing serum M2BP levels in each metabolic syndrome-related disease (obesity, hypertension, dyslipidemia, diabetes mellitus, and fatty liver) at baseline (Table 3A). Serum M2BP levels were higher in subjects with obesity, hypertension, dyslipidemia, diabetes mellitus, and fatty liver than in those without these diseases. We investigated which of these metabolic syndrome-related diseases were independent determinants for serum M2BP levels. Multiple logistic regression analyses indicated that obesity and fatty liver were independent determinants for serum M2BP levels (Table 3B). In male subjects, serum M2BP levels were higher in subjects with all metabolic syndrome-related diseases (Appendix A). In female subjects, serum M2BP levels were higher in subjects with obesity, hypertension, dyslipidemia, and fatty liver, but not with diabetes mellitus (Appendix A). Multiple logistic regression analyses indicated that obesity, hypertension, diabetes mellitus, and fatty liver were independent determinants for serum M2BP levels in male, and fatty liver was an only independent determinant for serum M2BP levels in female (Appendix A).

### 3.4. Serum M2BP Levels Were Significantly Correlated with BMI in FL+ Subjects but Not in FL− Subjects

Next, we investigated the relationships between serum M2BP levels and BMI in subjects with or without fatty liver disease (Figure 1). Serum M2BP levels were not correlated with BMI in FL− subjects, but were significantly correlated with BMI in FL+ subjects.

### 3.5. Relationships between Serum M2BP Levels at Baseline and Changes in Various Parameters

We investigated the relationships between baseline serum M2BP levels and changes in various parameters (Table 4). Baseline M2BP levels were significantly and negatively correlated with changes in CHE and LDL-C, which are produced in the liver and represent hepatic synthetic capacity. Thus, higher serum M2BP levels would predict reduced hepatic synthetic capacity. We also observed that baseline M2BP levels were significantly and negatively correlated with changes in platelet count, and significantly and positively correlated with changes in FIB4-index and NFS. This result indicated that higher serum M2BP level could predict liver fibrosis progression.

### 3.6. Serum M2BP Levels Could Serve as a Predictive Biomarker for Liver Fibrosis Progression

Serum M2BP levels were not significantly correlated with platelet count, FIB4-index, or NFS at baseline (Table 2). However, the four parameters of M2BP, platelet count, FIB4-index, and NFS are useful liver fibrosis markers for fatty liver disease [12,36,37,38]. We previously measured serum M2BP levels in 512 biopsy-proven NAFLD patients, and set a cut-off value (1.80 μg/mL) for the diagnosis of NASH from NAFLD [28]. In the present study, we compared the ability to predict changes in FIB4-index using the baseline values of FIB4-index, NFS, and serum M2BP levels (Table 5). Both FIB-4 index and NFS have two cut-off points (FIB4-index: low cut-off point, 1.30; high cut-off point, 2.67, NFS: low cut-off point, −1.455; high cut-off point, 0.676) [12,35]. We stratified study subjects based on low FIB4-index cut-off point (1.30), low NFS cut-off point (−1.455), or serum M2BP level (1.80 μg/mL), and investigated the relationships between these parameters and changes in the FIB4-index. The baseline FIB4-index and NFS were not correlated with changes in the FIB4-index. By contrast, baseline serum M2BP levels were significantly correlated with changes in the FIB4-index, and this correlation became stronger in subjects with higher baseline serum M2BP levels. These results indicated that subjects with higher serum M2BP levels at baseline would undergo liver fibrosis progression.

Decreased blood platelet count, FIB4-index, and NFS are useful clinical biomarkers for liver fibrosis. We performed multivariate analysis to identify baseline parameters that predicted changes in platelet count (Table 6A), FIB4-index (Table 6B), and NFS (Appendix A). We found that BMI and serum M2BP levels were significant negative determinants for platelet count changes, whereas serum TG and iron levels were significant positive determinants for platelet count changes. We also found that serum M2BP level was a significant positive determinant for FIB4-index changes, whereas serum TG was a significant negative determinant for FIB4-index changes. Serum M2BP level, age, ALT, albumin, TG, and HbA1c were significant positive determinants for NFS changes, whereas serum iron was a significant negative determinant for NFS changes (Appendix A).

### 3.7. Relationships between Changes in M2BP Levels and Other Parameters

Next, we investigated the relationships between changes in serum M2BP levels and other parameters (Table 7A). Changes in serum M2BP levels were significantly and positively correlated with changes in BMI, alcohol consumption, SBP, AST, ALT, GGT, CHE, TG, T-Chol, LDL-C, FBG, and HbA1c, whereas changes in HDL-C were significantly and negatively correlated with changes in serum M2BP levels. These results indicated that serum M2BP levels would increase with metabolic syndrome progression and liver dysfunction.

### 3.8. Changes in GGT and CHE Were Independent Determinants for Changes in M2BP Levels

Finally, we performed multivariate analyses to investigate which parameter changes could predict changes in serum M2BP levels (Table 7B). The results indicated that changes in SBP, GGT, CHE, and HbA1c were significantly and positively correlated with changes in serum M2BP levels. In male subjects, changes in SBP, ALT, and GGT were significantly and positively correlated with changes in serum M2BP levels (Appendix A). In female subjects, CHE was positively and ALT was negatively correlated with changes in serum M2BP levels (Appendix A). These effects of gender differences on changes in M2BP would need further investigation.

## 4. Discussion

We recently reported that M2BP was a useful biomarker that discriminated NAFLD from NASH [18,21,28,36]. Serum M2BP level is a useful predictive biomarker for NAFLD fibrosis progression, and M2BP levels increase during the progression of fibrosis. To investigate whether serum M2BP level can predict the development of liver fibrosis, we conducted a 7-year longitudinal study. There are four main results of our study: (1) serum M2BP levels at baseline were significantly correlated with changes in platelet count, FIB-4 index, and NFS, which are clinically used as liver fibrosis markers; (2) serum M2BP levels were most significantly correlated with fatty liver among the metabolic syndrome-related diseases; (3) serum M2BP levels were significantly correlated with BMI in subjects with fatty liver, but not in subjects without fatty liver; (4) changes in SBP, GGT, CHE, and HbA1c were independent determinants for changes in M2BP levels. These combined results indicated that serum M2BP level was closely associated with fatty liver disease and would serve as a predictive biomarker for liver fibrosis progression in patients with fatty liver disease.

Serum M2BP levels were not correlated with platelet count, FIB-4 index, and NFS at baseline. Our previous study demonstrated serum M2BP levels significantly correlated with FIB4-index and NFS [21,28]. In the present study, our study subjects were those who had received health check-ups and there would be few subjects with advanced liver fibrosis. The frequency of subjects whose value of FIB4-index was over 2.67 was 14.3% (73 among 512 NAFLD patients) in our previous study, while it was 3.1% (22 among 715 subjects) in this study at baseline. This difference in the study subjects between our previous study and present study would have any significance on these relationships between serum M2BP levels and liver fibrosis biomarkers. Platelet count decreases with liver fibrosis progression, and platelet count are often used as a clinical liver fibrosis marker. FIB4-index and NFS are scoring systems developed for the assessment of liver fibrosis progression [12,34,35]. Our longitudinal study demonstrated that baseline serum M2BP levels were significant determinants for platelet count decrease, FIB4-index increase, and NFS increase. Our present results indicated that baseline serum M2BP levels could predict liver fibrosis progression.

Recent studies demonstrated that a screening test using FIB4-index improved the management of patients with fatty liver [39,40]. Based on the results of our present study, we developed a two-step noninvasive protocol for the management of patients with metabolic fatty liver (Appendix A). Step 1 was a calculation of the FIB4-index. Subjects with FIB4-index <1.30 (424 subjects in this study) were considered as being at low risk of advanced liver fibrosis; they remained in clinical observation with re-assessment of advanced liver fibrosis risk after several years. Subjects with FIB-4 index >2.67 (22 subjects) were considered as high risk of advanced liver fibrosis, and were recommended for liver biopsy and detailed management by hepatologists. Step 2 was a measurement of serum M2BP levels. For the diagnosis of NASH from NAFLD, we set the cut-off value of serum M2BP as 1.80 μg/mL in our previous study [28]. Subjects with intermediate FIB4-index of 1.30–2.67 (269 subjects) received a test to measure serum M2BP levels; those with serum M2BP levels <1.80 μg/mL (179 subjects) remained under clinical observation, whereas those with serum M2BP levels >1.80 μg/mL (90 subjects) were recommended for detailed management by hepatologists. According to our protocol, about one-third of subjects with intermediate FIB4-index were recommended for detailed management.

Our previous study used various mouse liver disease models to analyze serum M2BP levels in liver fibrosis progression and liver inflammation [41]. M2BP belongs to the Scavenger receptor cysteine-rich domain (SRCR) superfamily of proteins involved in immune defense and regulation [42]. M2BP is expressed in many tissues and various types of cell including macrophages, hepatocytes, and hepatic stellate cells. Its expression in mouse macrophages can be upregulated by adherence and inflammatory cytokines (tumor necrosis factor-α (TNF-α) and interferon-γ (IFN-γ)) [43]. In chronic and acute liver injury mouse models, gene expression of M2BP significantly increased in liver [41]. Increased serum M2BP levels in liver diseases would mainly be produced from liver. Considering these findings, we believe that M2BP is associated with immunoreaction of liver fibrosis [23,25,27,44]. Although some differences in the molecular structures of human and mouse M2BP have been reported, M2BP-deficient mice exhibited a higher death rate after intraperitoneal lipopolysaccharide administration than wild-type mice [43]. This result indicated that inflammation-induced M2BP increase would have anti-inflammatory effects as a defensive substrate. In our study, subjects with higher M2BP levels would have enhanced liver inflammation, but the anti-inflammatory effects of M2BP would be not sufficient to suppress inflammation, which led to further liver fibrosis progression during the 7-year study. In addition, our study demonstrated fatty liver induced by aggravated nutritional status would have significant roles in changes in serum M2BP levels. Correcting nutritional status in fatty liver patients should reduce serum M2BP levels and lead to the improvement of fatty liver disease.

Our study has some limitations. First, the major limitation of these types of study is that only healthy subjects during 7-years tended to be analyzed. Second, we did not perform liver biopsy in our study subjects and could not directly examine whether serum M2BP levels predicted liver fibrosis progression. Our study subjects received health examinations, and liver biopsy was not feasible in this study for ethical reasons. Third, a follow-up period of 7 years might not be sufficient to investigate a possible role of M2BP in the progression of liver fibrosis. Forth, this is an observation study that cannot investigate the effects of interventional therapies on changes in serum M2BP levels. In future studies, we will resolve these limitations by including biopsy-proven liver disease patients. Fifth, we could not obtain data about menopause, we did not analyze the effects of menopause in female subjects. Our demonstrated data were sometimes different between male and female subjects. Female hormone would have a significance on these distinctions.

## 5. Conclusions

In conclusion, this is the first prospective observational study that demonstrates relationships between serum M2BP levels and other metabolic parameters. We found that serum M2BP levels increased in various metabolic syndrome-related diseases, especially in fatty liver disease. In addition, serum M2BP levels significantly associated with metabolic parameters, and nutritional status resulted in significant changes in serum M2BP levels. In addition, we found that serum M2BP levels at baseline could be used to predict changes in FIB4-index and NFS, which are useful liver fibrosis scoring systems. Our results indicate the importance of measuring serum M2BP levels in the management of patients with fatty liver disease.

## Figures and Tables

**Figure 1 nutrients-12-01770-f001:**
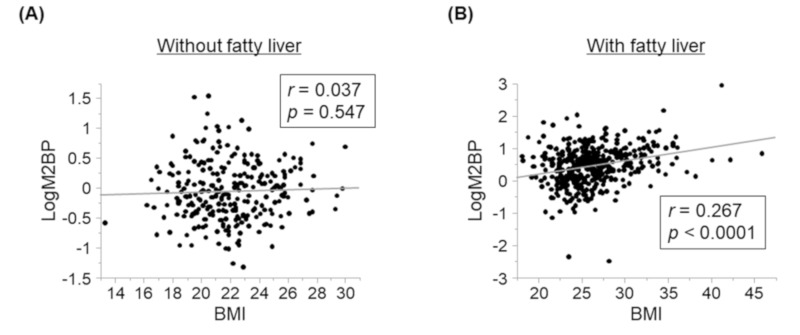
Relationships between serum M2BP levels and body mass index at baseline in subjects with or without fatty liver disease. (**A**) Relationships between serum M2BP levels and BMI in subjects without fatty liver. There were no significant relationships between serum M2BP levels and BMI in subjects without fatty liver. (**B**) Relationships between serum M2BP levels and BMI in subjects with fatty liver. There were significant positive relationships between serum M2BP levels and BMI in subjects with fatty liver.

**Table 1 nutrients-12-01770-t001:** Clinical and biochemical parameters of study subjects.

Variable	Baseline Parameters			Follow-Up Parameters after 7 Years		
	Fatty Liver (−)	Fatty Liver (+)	*p* Value *	Fatty Liver (–)	Fatty Liver (+)	*p* Value *
Number of study subjects	273	442		292	423	<0.0001 ^#^
Age (years)	54.0 ± 8.3	53.5 ± 6.8	N.S.	61.4 ± 8.7	59.8 ± 6.3	<0.05
Gender (M/F)	155/118	366/76	<0.0001	175/117	346/77	<0.0001
BMI (kg/m^2^)	22.0 ± 2.7	26.3 ± 3.8	<0.0005	21.9 ± 2.8	26.0 ± 3.8	<0.0001
Alcohol consumption (g/week)	86.6 ± 112.6	109.3 ± 128.5	N.S.	91.0 ± 123.5	114.1 ± 132.2	<0.05
SBP (mm Hg)	110.4 ± 14.3	120.0 ± 15.4	<0.0001	106.8 ± 15.8	114.2 ± 14.2	<0.0001
AST (U/L)	24.3 ± 13.1	32.1 ± 15.4	<0.0001	23.2 ± 10.2	30.4 ± 18.3	<0.0001
ALT (U/L)	23.4 ± 17.4	43.8 ± 24.8	<0.01	19.5 ± 10.0	35.7 ± 23.4	<0.0001
GGT (U/L)	50.8 ± 85.2	74.0 ± 78.7	<0.0001	42.3 ± 62.3	62.8 ± 66.5	<0.0001
T-Bil (mg/dL)	0.79 ± 0.32	0.80 ± 0.30	N.S.	0.80 ± 0.30	0.86 ± 0.34	<0.05
Albumin (g/dL)	4.23 ± 0.24	4.40 ± 0.22	<0.0001	4.33 ± 0.23	4.46 ± 0.23	<0.0001
Creatinine (mg/dL)	0.78 ± 0.16	0.83 ± 0.16	<0.0001	0.80 ± 0.18	0.87 ± 0.30	<0.0001
CHE (U/L)	314.7±62.0	378.7 ± 67.5	<0.0001	315.1 ± 63.9	358.9 ± 65.5	<0.0001
TG (mg/dL)	92.0 ± 83.3	157.7 ± 108.3	<0.0001	89.6 ± 50.4	143.5 ± 115.2	<0.0001
T-Chol (mg/dL)	199.1 ± 34.6	208.4 ± 34.0	<0.0001	202.1 ± 35.6	197.9 ± 33.3	N.S.
HDL-C (mg/dL)	65.5 ± 13.9	53.8 ± 10.7	<0.0001	67.9 ± 15.3	56.5 ± 12.3	<0.0001
LDL-C (mg/dL)	120.8 ± 33.0	138.7 ± 30.5	<0.0001	121.4 ± 30.7	125.7 ± 29.1	<0.05
Uric Acid (mg/dL)	5.30 ± 1.35	6.13 ± 1.32	<0.0001	5.27 ± 1.32	5.89 ± 1.26	<0.0001
FBG (mg/dL)	108.0 ± 27.9	120.4 ± 32.6	<0.0001	107.9 ± 22.7	120.7 ± 26.9	<0.0001
HbA1c (%)	6.09 ± 0.99	6.49 ± 1.08	<0.0001	6.14 ± 0.82	6.52 ± 0.99	<0.0001
Iron (µg/dL)	111.1 ± 38.2	117.1 ± 37.8	N.S.	112.8 ± 38.3	121.0 ± 38.3	<0.005
Platelet count (×10^4^/μL)	21.4 ± 4.9	21.8 ± 5.0	N.S.	21.3 ± 5.3	21.0 ± 4.9	N.S.
FIB4-index	1.39 ± 0.6	1.30 ± 0.61	<0.05	1.67 ± 0.82	1.61±1.03	<0.05
NFS	−1.32 ± 1.04	−1.31 ± 1.07	N.S.	−1.23 ± 1.17	−1.02 ± 1.13	<0.05
M2BP (μg/mL)	1.06 ± 0.61	1.85 ± 1.34	<0.0001	1.12 ± 0.95	1.61 ±1.30	<0.0001

Values represent mean ± SD; * *p* values correspond to the comparison between groups without and with fatty liver disease; Wilcoxon’s test for continuous factors and Pearson’s chi-square test for categorical factors were used; ^#^ Pearson’s chi-square test between data at baseline and follow-up. N.S.; not significant. Abbreviations: BMI, body mass index; SBP, systolic blood pressure; AST, aspartate aminotransferase; ALT, alanine aminotransferase; GGT, gamma glutamyltransferase; T-Bil, total bilirubin; CHE, choline esterase; TG, triglyceride; T-Chol, total cholesterol; HDL-C, high density lipoprotein cholesterol; LDL-C, low density lipoprotein cholesterol; FBG, fasting blood glucose; FIB4-index, Fibrosis-4 index; NFS, NAFLD fibrosis score; M2BP, Mac-2 binding protein.

**Table 2 nutrients-12-01770-t002:** Correlation between serum M2BP levels and various parameters at baseline.

Variable	M2BP	
	*r*	*p* Value
Age (years old)	0.023	N.S.
BMI (kg/m^2^)	0.374	<0.0001
Alcohol consumption (g/week)	0.018	N.S.
SBP (mm Hg)	0.233	<0.0001
AST (U/L)	0.328	<0.0001
ALT (U/L)	0.351	<0.0001
GGT (U/L)	0.279	<0.0001
T-Bil (mg/dL)	−0.105	<0.01
Albumin (mg/dL)	0.085	<0.05
Creatinine (mg/dL)	−0.045	N.S.
CHE (U/L)	0.303	<0.0001
TG (mg/dL)	0.326	<0.0001
T-Chol (mg/dL)	0.113	<0.005
HDL-C (mg/dL)	−0.191	<0.0001
LDL-C (mg/dL)	0.132	<0.0005
Uric acid (mg/dL)	0.115	<0.005
FBG (mg/dL)	0.099	<0.05
HbA1c (%)	0.119	<0.005
Iron (μg/dL)	−0.002	N.S.
Platelet count (×10^4^/μL)	0.043	N.S.
FIB4-index	0.061	N.S.
NFS	0.058	N.S.

Abbreviations are defined in the notes for Table 1.

**Table 3 nutrients-12-01770-t003:** Relationship between serum M2BP levels and metabolic risk factors. (**A**) Comparisons between serum M2BP levels and each metabolic syndrome–related disease at baseline. (**B**) Multiple logistic regression analysis of factors associated with serum M2BP levels at baseline.

(A)				
**Disease**	**Positive**	**Negative**	***p*** **Value**	
Obesity	1.88 ± 1.38	1.29 ± 0.92	<0.0001	
Hypertension	1.80 ± 1.10	1.41 ± 1.20	<0.0001	
Dyslipidemia	1.77 ± 1.11	1.40 ± 1.20	<0.0001	
Diabetes mellitus	1.62 ± 1.30	1.43 ± 0.95	<0.05	
Fatty liver	1.85 ± 1.34	1.06 ± 0.61	<0.0001	
**(B)**				
**Factor**	***t*** **Value**	***p*** **Value**	**95% CI**	
			**Lower**	**Upper**
Obesity (*y* = 1, *n* = 2)	3	<0.005	0.0496	0.237
Hypertension (*y* = 1, *n* = 2)	1.64	N.S.	−0.0150	0.168
Dyslipidemia (*y* = 1, *n* = 2)	0.4	N.S.	−0.0732	0.110
Diabetes mellitus (*y* = 1, *n* = 2)	0.95	N.S.	−0.0444	0.127
Fatty liver (*y* = 1, *n* = 2)	5.83	<0.0001	0.197	0.396

Abbreviations are defined in the notes for Table 1.

**Table 4 nutrients-12-01770-t004:** Correlations between baseline serum M2BP levels and changes in various parameters.

	M2BP	
Variable	*r*	*p* Value
ΔBMI (kg/m^2^)	−0.094	<0.05
Δalcohol consumption (g/week)	0.031	N.S.
ΔSBP (mm Hg)	−0.087	<0.05
ΔAST (U/L)	0.012	N.S.
ΔALT (U/L)	−0.069	N.S.
ΔGGT (U/L)	−0.032	N.S.
ΔT-Bil (mg/dL)	0.111	N.S.
ΔAlbumin (mg/dL)	−0.042	N.S.
ΔCreatinine (mg/dL)	0.013	N.S.
ΔCHE (U/L)	−0.196	<0.0001
ΔTG (mg/dL)	−0.157	N.S.
ΔT-Chol (mg/dL)	−0.187	<0.05
ΔHDL-C (mg/dL)	0.001	N.S.
ΔLDL-C (mg/dL)	−0.152	<0.001
ΔUric acid (mg/dL)	−0.080	N.S.
ΔFBG (mg/dL)	0.039	N.S.
ΔHbA1c (%)	−0.0056	N.S.
ΔIron (μg/dL)	0	N.S.
ΔPlatelet count (×10^4^/μL)	−0.19	<0.05
ΔFIB4-index	0.2	<0.0001
ΔNFS	0.18	<0.0001

Abbreviations are defined in the notes for Table 1; Δ, change in each parameter at 7-year follow-up.

**Table 5 nutrients-12-01770-t005:** Correlations between FIB4-index, NFS, or serum M2BP levels at baseline and changes in FIB4-index divided by cut-off values of FIB4-index, NFS, or serum M2BP levels.

	FIB4-Index				NFS				M2BP			
	< 1.30 (*n* = 424)		1.30 < (*n* = 291)		< −1.455 (*n* = 303)		−1.455 < (*n* = 379)		<1.80 μg/Ml (*n* = 510)		1.80 μg/mL < (*n* = 205)	
	*r*	*p* Value	*R*	*p* Value	*r*	*p* Value	*r*	*p* Value	*r*	*p* Value	*r*	*p* Value
**ΔFIB4-index**	0.048	N.S.	0.096	N.S.	0.099	N.S.	0.16	<0.005	0.111	<0.05	0.309	<0.0001

Abbreviations are defined in the notes for Table 1.

**Table 6 nutrients-12-01770-t006:** M2BP could be a predictive biomarker for liver fibrosis progression. (**A**) Multivariate analysis of predicted changes in platelet count using baseline various variables. (**B**) Multivariate analysis of predicted changes in FIB4-index using baseline various variables.

(A)				
Variable	*t* Value	*p* Value	95% CI	
			Lower	Upper
Gender (F)	0.93	N.S.	−0.232	0.645
Age	−1.48	N.S.	−0.06942	0.00980
BMI (kg/m^2^)	−2.01	<0.05	−0.170	−0.0018
Alcohol consumption (g/week)	−1.24	N.S.	−0.0042	0.000948
SBP (mm Hg)	−0.34	N.S.	−0.0227	0.0160
ALT (U/L)	−0.71	N.S.	−0.0195	0.00918
GGT (U/L)	−0.72	N.S.	−0.00553	0.00258
T-Bil (mg/dL)	0.57	N.S.	−0.703	1.28
Albumin (mg/dL)	−0.99	N.S.	−1.83	0.605
Creatinine (mg/dL)	1.5	N.S.	−0.546	4.05
CHE (U/L)	−0.43	N.S.	−0.00569	0.00365
TG (mg/dL)	2.94	<0.005	0.00151	0.00756
T-Chol (mg/dL)	−1.61	N.S.	−0.0155	0.00154
Uric acid (mg/dL)	−0.56	N.S.	−0.322	0.179
Iron (μg/dL)	2.39	<0.05	0.00171	0.0176
FBG (mg/dL)	1.86	N.S.	−0.00091	0.0327
HbA1c (%)	−0.93	N.S.	−0.751	0.269
M2BP (µg/mL)	−2.93	<0.005	−0.628	−0.124
**(B)**				
**Variable**	***t*** **Value**	***p*** **Value**	**95% CI**	
			**Lower**	**Upper**
Gender (F)	−1.75	N.S.	−0.151	0.00889
Age	1.98	N.S.	5.51 × 10^−5^	0.0145
BMI (kg/m^2^)	0.75	N.S.	−0.00954	0.0213
Alcohol consumption (g/week)	0.62	N.S.	−0.00032	0.000619
SBP (mm Hg)	−0.86	N.S.	−0.00509	0.00200
ALT (U/L)	−1.43	N.S.	−0.00454	0.00071
GGT (U/L)	1.19	N.S.	−0.00029	0.00119
T-Bil (mg/dL)	1.12	N.S.	−0.0776	0.286
Albumin (mg/dL)	−0.92	N.S.	−0.326	0.119
Creatinine (mg/dL)	−1.35	N.S.	−0.709	0.132
CHE (U/L)	−1.23	N.S.	−0.00139	0.00032
TG (mg/dL)	−3.46	<0.001	−0.00153	−0.00042
T-Chol (mg/dL)	−0.62	N.S.	−0.00205	0.00107
Uric acid (mg/dL)	0.97	N.S.	−0.0232	0.0685
Iron (μg/dL)	−1.07	N.S.	−0.00224	0.000661
FBG (mg/dL)	0.25	N.S.	−0.00269	0.00346
HbA1c (%)	0.03	N.S.	−0.0921	0.0944
M2BP (µg/mL)	11.89	<0.0001	0.233	0.325

Abbreviations are defined in the notes for Table 1; CI, confidence interval.

**Table 7 nutrients-12-01770-t007:** Changes in serum M2BP levels were correlated with changes in parameters related to metabolic risk. (**A**) Correlation coefficients for changes in M2BP levels and changes in various variables. (**B**) Multivariate analysis of predicted changes in serum M2BP levels using changes in various variables.

(A)				
Variable		ΔM2BP		
		*r*		*p* Value
ΔBMI (kg/m^2^)		0.136		<0.001
Δalcohol consumption (g/week)		0.082		<0.05
ΔSBP (mm Hg)		0.114		<0.005
ΔAST (U/L)		0.118		<0.005
ΔALT (U/L)		0.089		<0.05
ΔGGT (U/L)		0.153		<0.0005
ΔAlbumin (mg/dL)		0.01		N.S.
ΔCHE (U/L)		0.183		<0.001
ΔTG (mg/dL)		0.1		<0.05
ΔT-Chol (mg/dL)		0.133		<0.0005
ΔHDL-C (mg/dL)		−0.11		<0.01
ΔLDL-C (mg/dL)		0.135		<0.001
ΔUric acid (mg/dL)		−0.016		N.S.
ΔCreatinine (mg/dL)		−0.014		N.S.
ΔFBG (mg/dL)		0.14		<0.001
ΔHbA1c (%)		0.177		<0.001
ΔIron (μg/dL)		−0.046		N.S.
ΔPlatelet count (×10^4^/μL)		−0.046		N.S.
ΔFIB4-index		−0.051		N.S.
ΔNFS		−0.039		N.S.
**(B)**				
**Variable**	***t*** **Value**	***p*** **Value**	**95% CI**	
			**Lower**	**Upper**
ΔBMI (kg/m^2^)	1.04	N.S.	−0.0290	0.0942
Δalcohol consumption (g/week)	0.47	N.S.	−8.89 × 10^−4^	0.00146
ΔSBP (mm Hg)	2.32	<0.05	0.00105	0.0126
ΔALT (U/L)	−0.43	N.S.	−0.005323	0.00342
ΔGGT (U/L)	2.69	<0.01	0.000603	0.00386
ΔAlbumin (mg/dL)	0.47	N.S.	−0.345	0.562
ΔCHE (U/L)	2.27	<0.05	0.000371	0.00516
ΔTG (mg/dL)	0.18	N.S.	−0.00101	0.00122
ΔT-Chol (mg/dL)	1.31	N.S.	−0.000987	0.00494
ΔUric acid (mg/dL)	−1.27	N.S.	−0.147	0.0315
ΔCreatinine (mg/dL)	1.36	N.S.	−0.132	0.725
ΔFBG (mg/dL)	−0.07	N.S.	−0.00489	0.00453
ΔHbA1c (%)	2.37	<0.05	0.0314	0.335
ΔIron (μg/dL)	−1.94	N.S.	−0.00382	2.63 × 10^−5^
ΔPlatelet count (×10^4^/μL)	0.78	N.S.	−0.0156	0.0360

Abbreviations are defined in the notes for Table 1.

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
