# Peer review of "Serum Mac-2 Binding Protein Levels Associate with Metabolic Parameters and Predict Liver Fibrosis Progression in Subjects with Fatty Liver Disease: A 7-Year Longitudinal Study"

_nutrients, 2020, doi:10.3390/nu12061770_

Round 1

Reviewer 1 Report

In this paper the authors offer a comprehensive analysis of the potential biomarker M2BP in a cohort of patients with fatty liver disease.

I would like the authors to address the following comments upon revision of their manuscript:

Abstract

The listing of the results in the central paragraph is confusing. When the authors speak about metabolic parameters should specify them better. It would help having the concepts summed-up in a shorter manner with some actual data to help the reader understand what 'significantly correlates' or 'strongly correlates' looks like.

Introduction

  1. when listing the various ways to diagnose liver fibrosis the authors do not cite other forms of non-invasive assessment methods, such as Fibroscan, or scores based on circulating factors, such as the MELD score. It would be important to introduce all methods to detect and monitor liver fibrosis at this stage, and discuss why there is still the need for a reliable biomarker (e.g. technology is expensive, data difficult to interpret, requirement for high-level expertise and resources, etc).
  2. the authors seem to focus their attention on NAFLD (non-alcoholic fatty liver disease) in the introduction. However, they include in the analysis also patients with a history of alcohol abuse. Despite I agree that patients with alcohol abuse develop fatty liver, I think that the authors should offer a broader introduction on fatty liver disease of various aetiologies if they are to include the alcoholic liver disease patients. 

Materials and Methods

  1. The authors mention 'substance abuse' as one of the exclusion criteria. However, they include patients with alcoholic liver disease, hence patients that do have liver injury as a consequence of substance abuse. It is not really an issue on the conceptual point of view, given that, as mentioned above, these patients do develop fatty liver. It is an issue on the methodological point of view, and the authors should resolve it.

Results

  1. Paragraph 3.2: there is a negative correlation between M2BP and TBilirubin that is not reported in the text and it is not commented on. I I would invite the authors to report it and add a short comment on its significance.
  2. The R in figure 1B is not very strong. The correlation may be skewed by the few patients that have very high BMI and M2BP levels. Could the authors comment/discuss the issue and verify whether the correlation holds significance if those few subject are not in the analysis?
  3. Table 4. I believe the conclusions the authors come to are an overstatement. They claim that M2BP is predictive of liver function decline. However, albumin change does not significantly correlate with M2BP levels. The authors also claim that M2BP correlates with progression of liver fibrosis. However, they only provides FIB4-index as a measurement. The authors should use other more up-to-date and/or sensitive measurements of liver fibrosis, such as MELD score, fibroscan, Nordic biomarkers, ELF score, pro-collagen 3 and collagen 3. I appreciate not all of them can be performed now that the study has concluded, however the authors should at least measure other serum biomarkers of liver fibrosis, and correlate them with M2BP levels. This is paramount to strengthen the study and justify the statements made by the authors about the predictive value of M2BP for liver function and fibrosis.
  4. For Table 5 and Table 6 the same issues with FIB4 index applies. The authors should repeat the analysis using other serum biomarkers predictive of liver fibrosis. Moreover, the authors should state better whether any of the patients is on anticoagulant therapy, which could affect platelets measurement. 
  5. Paragraph 3.7: BMI and cholesterol (LDL and HDL) measurements are only one of the parameters utilised for the diagnosis and monitoring of the metabolic syndrome. Therefore the conclusion of the authors around the predictive value of M2BP for the progression of the metabolic syndrome should be tuned down in the revised version of the manuscript. Otherwise, they should prove correlation with other factors, such as hyperglycemia/diabetes mellitus.
  6. How do the authors interpret the lack of correlation between changes in M2BP levels and FIB4 index and platelets? In light of the previous results, the authors should discuss the significance of this lack of correlation further,

Discussion

  1. The discussion is long and convoluted. The authors should shorten the part of re-cap of their data and sum up the main points of discussion in a concise manners. For example the re-cap of data starting at line 307 could be brought forward and combined with the sum-up of other data. 
  2. The authors still do not comment on the inclusion of patients with history of alcohol abuse, which should be further considered
  3. For the methodological reasons listed above, the conclusions are overstated, especially the one around the predictive value of M2BP for liver fibrosis progression.
  4. The authors have a significant bigger proportion of males than females in their study. They do not discuss the potential for this gender bias to skew the results, and I would suggest them to do so. Also, it would be ideal if all the multivariate analysis could be repeated by gender, to verify whether it has a significant impact.
  5. The paragraph starting at line 295 is very debatable. Patients with chronic liver disease experience systemic inflammation, and the increased levels of M2BP may not necessarily be the results of activation of liver macrophages. Other macrophages resident in other organs potentially involved during chronic liver disease (e.g. kidneys) may participate to its production. It would be interesting if the authors could discuss this point more in depth. 

Author Response

Response to the comments of reviewers

We thank the editor and reviewers for the positive assessment of our manuscript and for identifying areas that required corrections and/or modification. The red text in the revised manuscript is the corrected/modified text. All line numbers mentioned in each response to the comments refer to the small numbers that appear in the left margin of the text of the revised manuscript.

Reviewers' Comments to Author:

Reviewer: 1

Comments and Suggestions for Authors

In this paper the authors offer a comprehensive analysis of the potential biomarker M2BP in a cohort of patients with fatty liver disease. I would like the authors to address the following comments upon revision of their manuscript:

Abstract

The listing of the results in the central paragraph is confusing. When the authors speak about metabolic parameters should specify them better. It would help having the concepts summed-up in a shorter manner with some actual data to help the reader understand what 'significantly correlates' or 'strongly correlates' looks like.

Thank you for the reviewer’s comments. According to the reviewer’s comments, we modified the descriptions in abstract of our revised manuscript (page 1, line 25-34).

Introduction

  1. when listing the various ways to diagnose liver fibrosis the authors do not cite other forms of non-invasive assessment methods, such as Fibroscan, or scores based on circulating factors, such as the MELD score. It would be important to introduce all methods to detect and monitor liver fibrosis at this stage, and discuss why there is still the need for a reliable biomarker (e.g. technology is expensive, data difficult to interpret, requirement for high level expertise and resources, etc).

Thank you for the reviewer’s important suggestion. We added some descriptions about non-invasive assessment methods for liver fibrosis in Introduction (page 2, line 49-55).

  1. the authors seem to focus their attention on NAFLD (nonalcoholicfatty liver disease) in the introduction. However, they include in the analysis also patients with a history of alcohol abuse. Despite I agree that patients with alcohol abuse develop fatty liver, I think that the authors should offer a broader introduction on fatty liver disease of various aetiologies if they are to include the alcoholic liver disease patients.

Thank you for the reviewer’s critical comments. As the reviewer pointed out, we modified our manuscript. In Introduction, we described about the criteria of MAFLD regardless of alcohol consumption (page 2, line 81-87), and added some descriptions in Patients and Methods (page 3, line 103-106).

Materials and Methods

  1. The authors mention 'substance abuse' as one of the exclusion criteria. However, they include patients with alcoholic liver disease, hence patients that do have liver injury as a consequence of substance abuse. It is not really an issue on the conceptual point of view, given that, as mentioned above, these patients do develop fatty liver. It is an issue on the methodological point of view, and the authors should resolve it.

Thank you for the reviewer’s important comments. According to the reviewer’s comments, we modified some descriptions in Patients and Methods (page 3, line 103-106), and added some analysis including alcohol consumption data (Table 1, 2, 4, 6, 7).

Results

  1. Paragraph 3.2: there is a negative correlation between M2BP and T Bilirubin that is not reported in the text and itis not commented on. I would invite the authors to report it and add a short comment on its significance.

We added short descriptions in our revised manuscript (page 5, line 178). Thank you.

  1. The R in figure 1B is not very strong. The correlation maybe skewed by the few patients that have very high BMI and M2BP levels. Could the authors comment/discuss the issue and verify whether the correlation holds significance if those few subject are not in the analysis?

The reviewer’s comments were very important. We analyzed the relationships between BMI and M2BP concentrations in subjects without morbid obesity (BMI>35) as following (Appendix Table 1). As this table represented, the R value (R=0.232) decreased compared with that of all subjects with fatty liver (R=0.267), but the relationships were still significant.

Appendix Table 1. The relationships between BMI and M2BP concentrations in subjects without morbid obesity.

n

R

P value

FL(-)

265

0.037

N.S.

FL(+)

422

0.232

<0.0001

  1. Table 4. I believe the conclusions the authors come to are an overstatement. They claim that M2BP is predictive of liver function decline. However, albumin change does not significantly correlate with M2BP levels. The authors also claim that M2BP correlates with progression of liver fibrosis. However, they only provides FIB4-index as a/measurement. The authors should use other more up-to date and/or sensitive measurements of liver fibrosis, such as MELD score, fibroscan, Nordic biomarkers, ELF score, pro-collagen 3 and collagen 3. I appreciate not all of them can be performed now that the study has concluded, however the authors should at least measure other serum biomarkers of liver fibrosis, and correlate them with M2BP levels. This is paramount to strengthen the study and justify the statements made by the authors about the predictive value of M2BP for liver function and fibrosis.

Thank you for the reviewer’s valuable comments. As the reviewer pointed out, changes in serum albumin levels did not correlated with serum M2BP levels at baseline. In NAFLD patients, serum albumin levels are hard to decrease till advanced liver cirrhosis. I think there should be few advanced liver cirrhosis patients in this study. We modified the description of our revised manuscript (page 8, line 220-221).

In the present study, we investigated the data of subjects receiving health check-ups. Therefore, our utilizable data were limited. Among non-invasive liver fibrosis assessment methods, we could not obtain Fibroscan, MELD score (our data lacked PT-INR), and other fibrosis biomarkers (hyaluronate, type 4 collagen 7S etc.). In our revised manuscript, we calculated NAFLD fibrosis score (NFS), and added NFS to our analysis (Table 1, 2, 4, 5, 7A, S2). In our previous study, we investigated serum M2BP levels in 512 biopsy-proven NAFLD patients (Kamada Y et al. Hepatology 2015). In the previous study, serum M2BP levels significantly correlated with serum hyaluronate (HA) and type 4 collagen 7S (T4C7S), which are clinically used liver fibrosis biomarkers (Appendix Table 2). Serum HA and T4C7S levels increased in patients with advanced liver fibrosis. In contrast, serum M2BP levels demonstrated significant stepwise increases in biopsy-proven NAFLD patients. Although, we could not perform liver biopsy, we think there were few subjects with advanced liver fibrosis in our present study.

Appendix Table 2. The correlation between serum M2BP levels and liver fibrosis biomarkers (hyaluronate, type 4 collagen 7S) in biopsy-proven NAFLD patients (n=512).

R

P value

HA

0.244

<0.01

T4C7S

0.174

<0.01

  1. For Table 5 and Table 6 the same issues with FIB4 index applies. The authors should repeat the analysis using other serum biomarkers predictive of liver fibrosis. Moreover, the authors should state better whether any of the patients is on anticoagulant therapy, which could affect platelets measurement.

Thank you for the reviewer’s valuable comments. In this study, we investigated the data of subjects receiving health check-ups. Therefore, our utilizable data were limited. Among non-invasive liver fibrosis assessment methods, we could not obtain Fibroscan, MELD score (our data lacked PT-INR), and other fibrosis biomarkers (hyaluronate, type 4 collagen 7S etc.). In our revised manuscript, we calculated NAFLD fibrosis score (NFS), and added NFS to our analysis (Table 1, 2, 4, 5, 7A, S2).

In addition, we added the following descriptions in our revised manuscript. Subjects receiving anticoagulant therapy, which could affect platelet measurements, were also excluded (page 3, line 105-106).

  1. Paragraph 3.7: BMI and cholesterol (LDL and HDL) measurements are only one of the parameters utilised for the diagnosis and monitoring of the metabolic syndrome. Therefore the conclusion of the authors around the predictive value of M2BP for the progression of the metabolic syndrome should be tuned down in the revised version of the manuscript. Otherwise, they should prove correlation with other factors, such as hyperglycemia/diabetes mellitus.

In paragraph 3.7 (Table 7A), changes in serum M2BP levels were significantly and positively correlated with changes in BMI, SBP, AST, GGT, CHE, T-Chol, LDL-C, FBG, and HbA1c, whereas changes in HDL-C were significantly and negatively correlated with changes in serum M2BP levels. We think these results indicated liver injury, hypercholesterolemia, and diabetes mellitus would have some significance with serum M2BP level increases.

  1. How do the authors interpret the lack of correlation between changes in M2BP levels and FIB4 index and platelets? In light of the previous results, the authors should discuss the significance of this lack of correlation further.

In our previous study, we investigated serum M2BP levels in 512 biopsy-proven NAFLD patients (Kamada Y et al. Hepatology 2015). In the previous study, serum M2BP levels significantly correlated with FIB4-index and NFS as following (Appendix Table 3). Serum M2BP levels demonstrated significant stepwise increases in biopsy-proven NAFLD patients. In the present study, our study subjects were health check-up received people. We could not perform liver biopsy, but we think there were few subjects with advanced liver fibrosis. FIB4-index and NFS in NAFLD patients tended to increase in patients with advanced liver fibrosis in our previous study. These discrepancies between previous study and present study would be important in the relationships between serum M2BP levels and scoring systems including FIB4-index and NFS. We added some descriptions in our revised manuscript (page 13, line 301-306).

Appendix Table 3. Correlation between serum M2BP and scoring systems (FIB4-index, NFS) in biopsy-proven NAFLD patients (n=512).

R

P value

FIB4-index

0.263

<0.0001

NFS

0.223

<0.01

Discussion

  1. The discussion is long and convoluted. The authors should shorten the part of re-cap of their data and sum up the main points of discussion in a concise manner. For example the re-cap of data starting at line 307 could be brought forward and combined with the sum-up of other data.

Thank you for the reviewer’s precise comments. According to the reviewer’s comments, we modified the descriptions of Discussion in our revised manuscript (page, 13, line 325-343).

  1. The authors still do not comment on the inclusion of patients with history of alcohol abuse, which should be further considered.

Thank you for the reviewer’s important comments. According to the reviewer’s comments, we modified some descriptions in Patients and Methods (page 3, line 103-104), and added some analysis including alcohol consumption data (Table 1, 2, 4, 6, 7).

  1. For the methodological reasons listed above, the conclusions are overstated, especially the one around the predictive value of M2BP for liver fibrosis progression.

According to the reviewer’s comments, we modified some descriptions in Conclusions of our revised manuscript (page, 14, line 357-359)

  1. The authors have a significant bigger proportion of males than females in their study. They do not discuss the potential for this gender bias to skew the results, and I would suggest them to do so. Also, it would be ideal if all the multivariate analysis could be repeated by gender, to verify whether it has a significant impact.

Thank you for the reviewer’s fruitful comments. According to the reviewer’s comments, we repeated our multivariate analysis by gender (Table 3B, Table 7B in original manuscript). We added some new tables in our revised manuscript as supplementary data (Table S1ABCD, S3AB). We added some descriptions about this analysis in our revised manuscript (page 11, line 275-279).

  1. The paragraph starting at line 295 is very debatable. Patients with chronic liver disease experience systemic inflammation, and the increased levels of M2BP may not necessarily be the results of activation of liver macrophages. Other macrophages resident in other organs potentially involved during chronic liver disease (e.g. kidneys) may participate to its production. It would be interesting if the authors could discuss this point more in depth.

Thank you for the reviewer’s valuable comments. As reviewer suggested M2BP is produced from various organs and cells. In our present study using mouse liver disease models, we checked gene expression of several organs including kidney, liver, spleen, heart, and fat. We confirmed only hepatic gene expression of M2BP increased in liver injury models. We modified several descriptions in our revised manuscript (page, 13, line 325-343). 

Reviewer 2 Report

The authors present data for M2BP as a biomarker for NAFLD. This is a follow up, longitudinal study that follows a baseline study 7 years earlier. This study is comparing M2BP levels to other measurements of liver damage.

The study is well-designed and conducted. One question I have is whether they have data on the degree of use of common drugs that can injure the liver, such as statins, niacin, acetaminophen, certain anti-hypertensives, anti-fungals, antibiotics. Their frequent use may increase liver damage in susceptible patients, and are an important consideration in studying liver disease biomarkers. It is noted in the manuscript that they are aware that alcohol consumption was not exculsionary.

In Figure A1 the use of Mac-2bp is not consistent with the rest of the article, nor with its legend. M2BP should be used here. Also, the legend should include the definition of M2BP as well as the other abbreviations in the figure.

Author Response

Response to the comments of reviewers

We thank the editor and reviewers for the positive assessment of our manuscript and for identifying areas that required corrections and/or modification. The red text in the revised manuscript is the corrected/modified text. All line numbers mentioned in each response to the comments refer to the small numbers that appear in the left margin of the text of the revised manuscript.

Reviewers' Comments to Author:

Reviewer: 2

The authors present data for M2BP as a biomarker for NAFLD. This is a follow up, longitudinal study that follows a baseline study 7 years earlier. This study is comparing M2BPlevels to other measurements of liver damage. The study is well-designed and conducted.

One question I have is whether they have data on the degree of use of common drugs that can injure the liver, such as statins, niacin, acetaminophen, certain anti-hypertensives, anti-fungals, antibiotics. Their frequent use may increase liver damage insusceptible patients, and are an important consideration in studying liver disease biomarkers. It is noted in the manuscript that they are aware that alcohol consumption was not exculsionary.

Thank you for your important comments. According to the reviewer’s comments, we compared AST and ALT values in study subjects with or without drug use (hypertension, dyslipidemia). As shown in the following table, there are no significant difference between drug use (+) and (-) subjects. We added some descriptions about this points in our revised manuscript (page 4, line 135-137).

Appendix Table 4. Comparisons of serum transaminase levels in subjects with or without usage of antihypertensive drug or dyslipidemia drug.

antihypertensive drug

Dyslipidemia drug

drug use

(+)

(-)

P value

(+)

(-)

P value

n

146

570

119

597

AST

29.2 ± 13.2

29.1 ± 15.4

N.S.

28.2 ± 12.7

29.3 ± 15.4

N.S.

ALT

36.8 ± 20.6

35.7 ± 25.2

N.S.

36.1 ± 21.1

35.8 ± 24.9

N.S.

In Figure A1 the use of Mac-2bp is not consistent with the rest of the article, nor with its legend. M2BP should be used here. Also, the legend should include the definition of M2BP as well as the other abbreviations in the figure.

Thank you for your comments. We corrected “Mac-2bp” to “M2BP” in our revised manuscript.

Round 2

Reviewer 1 Report

I appreciate the authors addressed all my main concerns. I am happy with the level of revision provided and with the quality of the manuscript in its revised form. I have no further comments.

Author Response

We sincerely appreciate for your valuable comments. Thank you.